# Ultra-Wideband Differential Line-to-Balanced Line Transitions for Super-High-Speed Digital Transmission

**DOI:** 10.3390/s22186873

**Published:** 2022-09-11

**Authors:** Byung Cheol Min, Gwan Hui Lee, Jung Seok Lee, Syifa Haunan Nashuha, Hyun Chul Choi, Kang Wook Kim

**Affiliations:** 1School of Electronic and Electrical Engineering, Kyungpook National University, Daegu 41566, Korea; 2Ulsan National Institute of Science and Technology, Ulsan 44919, Korea

**Keywords:** ultra-wideband technology, conformal mapping, planar transmission lines, differential line, balanced line, DL-to-BL transitions, high-speed digital circuit

## Abstract

A conventional differential line (DL), commonly used on typical digital circuit boards for transmitting high-speed digital data, has fundamental limitations on the maximum signal bandwidth (~10 GHz), mainly due to signal skew, multiple line coupling, and EM interference. Therefore, to support super-high-speed digital data transmission, especially for beyond 5G communications, a practical high-performance transmission structure for digital signals is required. Balanced lines (BLs) can transmit the differential signals with multiple advantages of ultra-wide bandwidth, common-mode rejection, reduced crosstalk, phase recovery, and skew reduction, which enable super-high-speed transmission. In order to utilize the BLs in the DL-based digital circuit, connecting structures between a DL and BLs are required, but the DL-to-BL transition structures dominate the operating bandwidth and signal properties. Therefore, in this paper, properties, and design methods for two ultra-wideband DL-to-BL transitions, i.e., DL-to-CPS (coplanar stripline) and DL-to-PSL (parallel stripline) transitions, are presented. Both implemented DL-to-CPS and DL-to-PSL transitions provide high-quality performance up to 40 GHz or higher, significantly enhancing the frequency bandwidth for the transmission of digital signals while providing compatibility with the DL-based PCBs. The fabricated DL-to-CPS transition performs well from DC to 40 GHz with an insertion loss of less than 0.86 dB and a return loss of more than 10 dB, and the fabricated DL-to-PSL transition also provides good performance from DC to 40 GHz, with an insertion loss of less than 1.34 dB and a return loss of more than 10 dB. Therefore, the proposed DL-to-BL transitions can be applied to achieve super-high-speed digital data transmission with over 40 GHz bandwidth, which is more than four times the bandwidth of the DL, supporting over 200 Gbps of digital data transmission on PCBs for the next generation of advanced communications.

## 1. Introduction

Nowadays, in the middle of the fourth industrial revolution, advanced technologies such as 5G communications, the Internet of Things (IoT), and artificial intelligence, based on cloud computing and high-performance computing, have revolutionized information technology, which demands a huge volume of digital data transmission over 100 Gbps on a fast time scale [1]. Digital data transmission, which uses a form of square wave consisting of the main signal pulse and many harmonics, requires multiple times of the main signal frequency through the transmission lines to maintain the integrity of the signal. Therefore, for high-speed digital transmission, the frequency bandwidth of the transmission lines should be very wide. To enhance the digital data transmission speed, the signal transmission structure should have an ultra-wide bandwidth.

A high-speed digital signal line can be exposed to lots of external noises of various frequencies while the digital signal propagates along the signal transmission structures. Therefore, differential signaling is typically used to mitigate the external noise and detect the low-voltage digital signal [2]. A differential line (DL) is the most commonly used transmission line for differential signaling. The DL consists of two conductors and a ground plane, which is essentially a form of two microstrip lines laid out in parallel. The typical characteristic line impedance of the DL is 100 Ω, which is twice the single-ended (microstrip) line impedance of 50 Ω. The DL is easily fabricated and has good accessibility for installation of the various chips, with the advantages of reduced noise effects and low voltage signal detection. That is why the DL has been widely adopted over the single-ended (microstrip) line for the high-speed signal lines in most of the digital circuits [3].

A practical high-speed circuit board is typically crowded with chips, interconnects, and signal lines, and so the signal lines connecting the chips and components tend to have bending structures or vertical via connections, which are exposed to interference by adjacent signal lines. In spite of the reduced noise sensitivity of the DL, in these complicated environments, it may suffer some serious problems such as signal reflection, crosstalk, and signal skew. When the frequency of a digital signal in DL passes ~10 GHz, the undesired phenomena begin to appreciably affect the signal quality [4,5,6,7,8]. In these cases, application of the multiple levels (over 4 levels) of the pulse amplitude modulation (PAM) scheme to the DL is often limited, causing the data rate to be limited to less than 20 Gbps with the DL [9]. Therefore, to achieve the super-high-speed digital transmission required for next-generation communications, a new digital transmission line with an ultra-wide frequency bandwidth, even with complex line structures, is highly desired.

Another type of digital transmission line, which can support differential signaling but is less frequently used, is a balanced line (BL). A BL consists of two parallel lines, and two popular types of planar BLs are a coplanar stripline (CPS) and a parallel stripline (PSL). The CPS is formed with two uniplanar parallel conductor lines located side by side and is usually used for balanced antenna baluns [10,11,12,13]. The characteristic line impedance of the CPS with typical PCB substrates and fabrication conditions is usually high (e.g., 120–200 Ω with the 10 mil Duroid 5880 substrate). On the other hand, the PSL is formed with two parallel lines located vertically, with a dielectric substrate in between the two lines. The characteristic line impedance of the PSL can be changed in a wide range (e.g., 30–130 Ω with the 10 mil Duroid 5880 substrate). The PSL is used for the power divider, coupler, filter, and balanced mixer [14,15].

Utilizing the BLs for differential signaling may provide many advantages in terms of ultra-wide bandwidth, common-mode rejection, phase recovery, reduced crosstalk, and skew reduction. The BLs with proper transitional structures can provide a frequency bandwidth of several tens of gigahertz (GHz). With the BLs, two conductor strips are strongly coupled with each other so that one conductor can be a ground line for the other one, which enables them to block out the common-mode noise. Additionally, the PSL, a vertically symmetric structure, does not generate line-length imbalance with bent lines, thus educing the skew problem. In addition, the two strongly coupled lines in a BL tend to self-recover the phase imbalance related to a skew. One of the primary reasons that BLs have not been widely used in digital circuit boards until now is the lack of practical transitional structures between the commonly used DL and the BLs. Until now, various transmission line structures for differential signaling were known [16], but only a few cases of transitional structures between the differential transmission lines were reported [17,18], and the optimal transitional structures between DL and BLs were not yet reported.

A transition is a connecting structure between two transmission lines. The overall frequency bandwidth is often determined by the connecting structure. To maximize the advantages of the corresponding transmission lines, a transition connecting the transmission lines with good performance is indispensable. These days, most high-speed digital chips and interfaces are designed to mate with the DL, and thus the DL is the most conveniently used in high-speed digital circuit boards. Therefore, connecting structures between the DL and BLs are required to utilize the advantages of BLs.

Previously, various planar transition structures with ultra-wideband performance were proposed by the authors’ group. A CPS-to-PSL transition was designed with a frequency range from 6.4 GHz to 40 GHz, tapering the line impedance from 147 Ω (and 120 Ω) CPS to 50 Ω PSL [19]. An asymmetric CPS-to-microstrip line (MSL) transition was designed with a frequency range from 6 GHz to 40 GHz, tapering line impedance from 147 Ω CPS to 50 Ω MSL [20]. A PSL-to-MSL transition was designed with a frequency range up to 40 GHz, maintaining the impedance at 50 Ω [21]. In Table 1, the characteristics and performances of the related transitions are compared.

Therefore, in this paper, two types of practical high-performance transitional structures from the DL to BLs for typical printed circuit boards are proposed, i.e., the DL-to-CPS transition and the DL-to-PSL transition. Analytical formulas for the characteristic line impedances of the cross-sections of the transition, based on conformal mapping, help to design the transitions efficiently and accurately with optimal performance. In the proposed transitions, the two signal lines of the DL are optimally converted into two signal lines of BLs either on the top or on the bottom of the substrate, and these transitions are compatible with DL-based circuit boards. Since these ultra-wideband DL-to-BL transitions provide bandwidth of several tens of GHz with skew reduction and self-phase recovery properties, these DL-to-BL transitions are quite adequate for the digital signal transmission line of the next generation super-high-speed circuit.

## 2. Differential Line and Balanced Lines for Digital Signaling

### 2.1. Differential Line and Its Issues

A DL is the most widely used transmission line for differential signaling on typical PCBs. Figure 1 shows a conventional DL structure, which is composed of two conductor signal lines, a ground plane, and a substrate in between them. The two conductor lines of opposite polarities with a gap are placed on the top side of the substrate. The DL is equivalent to two coupled microstrip lines of opposite polarities with a gap. The conductor lines are electromagnetically (EM) coupled, and the amount of the EM coupling mostly depends on the gap width between the lines.

A high-speed digital circuit board consists of lots of chips and signal lines laid out within a limited space, and so complex line structures, such as bending structures, adjacent multiple signal lines, and via connections to other layers, are inevitable, some of which are illustrated in Figure 2. The DL layout for a circuit board for high-speed digital signaling follows the design guidelines suggested in [22,23,24,25,26]. However, the proper amount of EM coupling by adjusting the gap width between two signal lines is still debated [27], and undesired phenomena can happen in such complex line structures.

Skew is one of the undesirable phenomena that can occur due to the line length imbalance of the DL, especially in the case of a bending structure. The skew not only distorts the digital signal shape but also generates a common-mode noise, which may cause EM interference affecting the other signal lines. To reduce the skew, an extra compensation structure to equalize the line length is typically required, as illustrated in Figure 3a, but the EM interference can be generated in the DL (with a length of *L*) before reaching the compensation structure. Figure 3b shows the phase difference between two lines of the DL after applying the compensation structure as a function of the distance *L* from the bending structure to the compensation structure. The phase difference between the two lines appreciably deviates from 180° at high frequencies (over 15 GHz), more significantly with the increased distance (e.g., a phase difference of 158° at 15 GHz with a distance of 1000 mil). In addition, a compensation structure with a very wide bandwidth for high-speed digital signals is very difficult to design. The bending structures using tight coupling and an additional structure could reduce the skew in [6,7,8], but the frequency bandwidths of the structures were insufficient for transmitting the high-speed signals. In fact, these phase differences of the imbalanced DL can be nonlinear in frequency as a function of the gap width, linewidth, line length, and substrate thickness of the DL, complicating the phase compensation with high-speed digital signals.

As another critical issue, crosstalk may cause a problem, especially with multiple line pairs lying in close proximity as shown in Figure 4. Each line of the DL can be exposed to the other adjacent signal lines at different distances, causing an imbalance in the differential signal and thus also generating the common-mode noise. In order to reduce the crosstalk, the DL with a tight gap is recommended in the industry [28]. However, the crosstalk levels for each line of the DL can be asymmetrical, causing an imbalance in the differential signal. A twisted DL structure was reported to reduce the crosstalk, but the structure may be inadequate to be applied to the complex high-speed circuit board due to the expensive fabrication cost [29].

Therefore, there exists a strong need for a new transmission line structure for differential signaling, overcoming the limits of the DL with high-speed signals, in the advent of the next generation of super-high-speed digital communications.

### 2.2. Balanced Lines: CPS and PSL

Two types of planar BLs for differential signaling, a CPS and a PSL, are shown in Figure 5. The CPS is a uniplanar structure with two conductor lines of the opposite polarities on the top side of the substrate, and the PSL is an antipodal structure in which a dielectric substrate is placed between the two conductor lines of the opposite polarities, as shown in Figure 5a,b, respectively. Each of the conductor lines of the BLs are considered a ground line to the other one, which supports the differential signaling. Additionally, thanks to structural symmetry, in the presence of curving corners, the PSL does not cause any line length imbalance between the two conductors, unlike the DL, and therefore does not cause skew with less layout area on PCBs than using the DL.

In the presence of multiple line pairs in close proximity, the PSL, one of the BLs, is less affected by the other PSL since the two lines of the PSL are symmetrically exposed to the electric fields from the other PSL, as shown in Figure 6a. With the DL, however, imbalance between the DL lines is unavoidable when the crosstalk occurs from the other DL [30] (as illustrated in Figure 4). Figure 6b shows the far-end crosstalk (FEXT) levels between the DLs and between the PSLs when the distance between the victim line and the aggressor line is 30 mil, which is three times the linewidth of the DL as following the design guideline in [31]. The crosstalk amplitudes of the two lines of the PSL are identical and similar to one of the DL lines that is located farther from the aggressor. Moreover, the difference in crosstalk levels on the two lines of the DL, which may produce the common-mode noise, is about 7.5 dB up to 40 GHz. Therefore, the PSL has an advantage in crosstalk levels as compared with the DL.

Another structural advantage of the BLs is that they can inherently obstruct the propagation of the common-mode signal or noise. With the common-mode signals, two signal lines of the BL possess the same polarity signals without their corresponding ground planes, and so most of the common-mode signals cannot propagate through the BL (but some portion of them can radiate along the lines of the BL). Using this intrinsic common-mode blocking property of the BLs, common-mode rejection filters were designed [32,33,34,35,36,37].

Additionally, an important property of the BLs is their capability to self-recover the phase imbalance when deviating from the opposite polarity. In the presence of a line length imbalance between the two conductor lines of the DL, the BLs connected to the DL have a tendency to align the signal to have the opposite polarities on the two lines as the signal propagates along the BL due to strong field coupling between the two conductors. For example, Figure 7 shows the phase differences of the two lines of the DL with the compensation structure and of the BLs with the optimal DL-to-BL transitions in the case of a 30 mil path difference. The phase-recovering property of the CPS and PSL with transitions recovers the balance of the differential signal, whereas the signal balance of the DL becomes broken at high frequencies (e.g., ~23 GHz).

Therefore, the BLs have multiple advantages in providing ultra-wide bandwidth, rejection of common-mode signals, skew reduction, reduced coupling between multiple lines, and phase recovery. With these advantages, especially in bending and crosstalk, the BLs can provide frequency bandwidth of several tens of GHz on complex digital circuit boards, supporting several tens to hundreds Gbps of digital data using the multi-level PAM modulation scheme [38]. When optimally connected with the DL, the BLs can expand the limited frequency range of the DL with their additional advantages, providing super-high-speed transmission. The key components in utilizing the BLs on DL-based circuit boards are the optimal transitional structures between the DL and BLs, and, in this paper, two types of optimal transition structures are proposed: DL-to-CPS and DL-to-PSL transitions.

## 3. Proposed DL-to-BL Transitions

Design methods for two types of DL-to-BL transitions are proposed, i.e., DL-to-CPS and DL-to-PSL transitions as shown in Figure 8a,b. Figure 8a illustrates a perspective view of the DL-to-CPS transition, consisting of two signal lines and a ground plane with an aperture. Figure 8b shows a perspective view of the DL-to-PSL transition, consisting of a DL-to-CPS transition and a CPS-to-PSL transition. To demonstrate the designs, two transitions are implemented with the Duroid 5880 substrate (relative permittivity *ε_r_* = 2.2) with a 10 mil (0.25 mm) thickness. The overall dimensions of the proposed transitions are 5.84 × 12.70 × 0.29 (mm) and 10.41 × 12.70 × 0.29 (mm), respectively.

## 4. Design of the DL-to-CPS Transition

### 4.1. Design

Figure 9 shows top and bottom views of the DL-to-CPS transition (in Figure 8a). The section from *AA**′* to *BB**′* is a conventional DL, where the linewidth and the gap width between the DL lines are notated as *w_d_* and *g_d_*, respectively. The transitional structure from the DL to the narrow-gap DL is the section from *BB**′* to *CC**′*. The linewidth of the top signal lines decreases from *w_d_* at *BB**′* to *w_m_* at *CC**′*, and the gap width decreases from *g_d_* at *BB**′* to *g_c_* at *CC**′*. The next section, from *CC**′* to *DD**′,* transforms the narrow-gap DL into the CPS, where a ground aperture begins to open wide. The gap between the lines on the top side is maintained as *g_c_*, while the linewidth of signal lines increases from *w_m_* at *CC**′* to *w_c_* at *DD**′*. As the ground aperture width increases, the transition structure forms a CPS line with a wide ground aperture. The section from *DD**′* to *EE**′* is a conventional CPS line. The dimensions of the design parameters (*w_d_*, *w_c_*, *w_m_*, *g_d_*, *g_c_*, and *l*) are listed in Table 2.

### 4.2. Electric Field Distribution

The electric field distributions at the cross-sectional stages of the DL-to-CPS transition are shown in Figure 10. The electric field distribution from *AA**′* to *BB**′* is identical to a conventional DL, which resembles the field distribution of two microstrip lines with opposite polarities and weak electric coupling between the top signal lines. The electric field intensity between the top signal lines increases along the transition structure from *BB**′* to *CC**′* as the signal linewidth and gap between the signal lines decrease. In the section from *CC**′* to *DD**′*, the narrow-gap DL is transformed into the CPS as the ground aperture flares out. Due to the aperture in the ground, the electric field is mainly concentrated in the gap area between the signal lines, and fringing fields from the signal lines to split ground lines are present. As the aperture becomes wider, the electric field lines from the signal-to-ground lines are reduced while the electric field intensity between the signal lines is increased. Finally, the electric field distribution of a conventional CPS line is formed when the ground aperture is very wide, as shown in the section from *DD**′* to *EE**′*.

### 4.3. Cross-Sectional Model and Analytical Formulas

Along the transition, the design parameters of the transitional cross-section, such as the linewidth *w* of the signal lines, the gap width *g* between the signal lines, and the aperture width *s* of the bottom ground line, are changed continuously. As the structural parameters are changed, the electric field distribution shape of the cross-section is also transformed. In designing the transition, the characteristic line impedance of the transitional cross-section is a critical design parameter.

In order to calculate the characteristic line impedance corresponding to the electric field distribution, a cross-sectional model can be used to derive analytical formulas of the line impedance at all locations of the transition. In Figure 11, a cross-sectional model with four analysis regions of the proposed DL-to-CPS transition is shown. Assuming the quasi-transverse electromagnetic (quasi-TEM) condition, this cross-sectional model can be analyzed using conformal mapping, which transforms the geometry of each region into a parallel plate structure [39]. Hence, the capacitance of each region can be obtained to calculate the characteristic line impedance of the whole cross-sections of the transition.


**Region I: Upper Region of the Transition**


Region I is equivalent to the upper plane of the CPS as shown in Figure 11a. The electric field of this region is formed as a CPS mode in the air. The capacitance of this analysis region is obtained by the following literature [39,40] as (1). *K* refers to the elliptical integral of the first kind, *ε*_0_ is the dielectric constant of air, and the modulus $k_{1}$ and the complementary modulus $k_{1}^{\prime }$ are the same as the moduli of Type 2 in [41].
(1)$$C_{1}=\varepsilon_{0}\frac{K\left(k_{1}^{\prime }\right)}{K\left(k_{1}\right)}$$


**Region II: Inside of the Substrate**


Region II is the inside region of the DL substrate with a ground aperture, and the region is symmetric with respect to an E-wall. Due to the symmetry of the structure, the Region II can be analyzed with only half of the region (Region II′), as shown in Figure 11b. Since the electric field lines connect the signal line to the E-wall and the ground line, the region can be divided into two analysis regions: the region between the signal line and the E-wall (Region II′(a)) and the region between the signal line and the ground line (Region II′(b)) as shown in Figure 11c and Figure 11d, respectively. The boundary between the two regions is determined by the width of the ground aperture.

Region II′(a) is half of the CPS inner region with relative permittivity *ε_r_*. The total capacitance of this region is obtained as (2). The modulus $k_{2a}$ and the complementary modulus $k_{2a}^{\prime }$ are equivalent to those of Type 2 in [41]. Region II′(b) is the inner region of a microstrip line with an asymmetric finite ground plane with relative permittivity *ε_r_*. The capacitance of this region can be obtained as (3). The modulus $k_{2b}$ and the complementary modulus $k_{2b}^{\prime }$ are equivalent to the moduli of Type 3 in [41].
(2)$$C_{2a}=\frac{\varepsilon_{0}\varepsilon_{r}}{2}\frac{K\left(k_{2a}^{\prime }\right)}{K\left(k_{2a}\right)}\;$$
(3)$$\;C_{2b}=\varepsilon_{0}\varepsilon_{r}\frac{K\left(k_{2b}^{\prime }\right)}{K\left(k_{2b}\right)}\;$$


**Region III: Upside Fringing Field**


Region III is the analysis region for fringing fields connecting the signal lines to the ground lines, where the electric field lines are mostly distributed in the air. Since the field lines connecting two signal lines and ground lines are symmetric, only half of the region (Region III′) is analyzed with respect to the E-wall. The capacitance of the divided region can be obtained as (4). The modulus $k_{3}$ and the complementary modulus $k_{3}^{\prime }$ are equivalent to the moduli of Type 5 in [41].
(4)$$C_{3}=\varepsilon_{0}\frac{K\left(k_{3}^{\prime }\right)}{K\left(k_{3}\right)}$$


**Region IV: Downside Fringing Field**


Region IV is the analysis region for fringing field lines through the ground aperture, where the electric field lines are also mostly distributed in the air, and half of the region with respect to the E-wall (Region IV′) must also be analyzed. The capacitance of the divided region can be obtained as (5). The modulus $k_{4}$ and the complementary modulus $k_{4}^{\prime }$ are equivalent to the moduli of Type 5 in [41].
(5)$$C_{4}=\varepsilon_{0}\frac{K\left(k_{4}^{\prime }\right)}{K\left(k_{4}\right)}$$


**Characteristic Line Impedance**


The characteristic line impedance and effective permittivity of the cross-sectional area of the proposed transition can be calculated by summing up the capacitances of the four regions, expressed as (6) and (7), respectively.
(6)$$Z_{0}=2\frac{120\pi \varepsilon_{0}}{\sqrt{\varepsilon_{eff}}\left(C_{1}/2+C_{2a}/\varepsilon_{r}+C_{2b}/\varepsilon_{r}+C_{3}+C_{4}\right)}$$
(7)$$\varepsilon_{eff}=\frac{\left(C_{1}/2+C_{2a}+C_{2b}+C_{3}+C_{4}\right)}{\left(C_{1}/2+C_{2a}/\varepsilon_{r}+C_{2b}/\varepsilon_{r}+C_{3}+C_{4}\right)}\;$$

### 4.4. Calculation of Characteristic Line Impedance

As an example of the proposed DL-to-CPS transition, a DL with an impedance of 100 Ω connects to the CPS with an impedance of 138 Ω. Since the gap width for a 100 Ω CPS with typical PCB substrates is too small, a 138 Ω line impedance is selected for the CPS line (*g_c_* = 5 mil (0.13 mm) with the 10 mil Duroid 5880 substrate). The calculated characteristic line impedance values using (6) are compared with the 3D EM-simulated values in Figure 12a,b.

Figure 12a represents a line impedance taper from the DL (100 Ω) to the narrow-gap DL (138 Ω) (from *AA**′* to *CC**′* in Figure 2) with the optimal Klopfenstein impedance taper as reducing the DL gap width and linewidth. It can be observed that the calculated impedance values deviate from the EM-simulated impedance values by less than 3.7%. This discrepancy may have been caused by an overlap between Regions I and III (shown in Figure 11a) in the case of a wide gap (for *g* > 13.2 mil).

Figure 12b shows the characteristic line impedance values for a narrow-gap DL-to-CPS transition (from *CC′* to *DD′* in Figure 2), maintaining the line impedance at 138 Ω. The calculated impedance values as a function of aperture width *s* deviate from the EM-simulated impedance values by less than 4.1%. The calculated impedance is slightly higher than the EM-simulated result, which can be attributed to insufficient treatment of the boundary between Regions II′(a) and II′(b) (shown in Figure 11c,d).

## 5. Design of the DL-to-PSL Transition

### 5.1. Design

Figure 13 shows top and bottom views of the proposed DL-to-PSL transition (in Figure 8b). The DL-to-PSL transition consists of a DL-to-CPS transition and a CPS-to-PSL transition. The section from *AA**′* to *DD**′* is the DL-to-CPS transition, which is identical to the transition described in Section 4. The next section from *DD**′* to *FF**′* is the CPS-to-PSL transition, which is a combination of a CPS-to-asymmetric PSL transition and an asymmetric PSL-to-PSL transition, is presented in [19]. At *DD**′, a* bottom signal line appears at *DD*, which is connected through the vias to one of the top signal lines. The other signal line’s linewidth decreases from *w_c_* at *DD**′* to *w_t_* at *EE**′*. In the section from EE′ to FF′, as the top signal line containing the vias is removed, an asymmetric PSL is formed. Linewidths of the top and the bottom signal lines change continuously until both lines reach the width of *w_p_* at *FF**′*. Finally, the section from *FF**′* to *GG**′* is a conventional PSL. The dimensions of the design parameters (*w_d_*, *w_c_*, *w_t_*, *w_p_*, *g_d_*, *g_c_*, and *l*) are listed in Table 3.

### 5.2. Electric Field Distribution

Figure 14 depicts the electric field distributions at all representative cross-sections of the transition. From *AA**′* to *DD**′*, the electric field distributions are the same as those of the DL-to-CPS transition. In the section from *DD**′* to *EE**′*, one of the CPS lines is connected to the bottom signal line through the vias. The electric field distribution consists of half of a CPS mode in the air, a CPS mode with a finite width of the bottom signal line in the substrate, and a fringing field from the top signal line to the bottom line. As the width of the bottom line is increased from *DD**′* to *EE**′*, the electric field inside the substrate is mostly distributed between the top and bottom signal lines, as shown in the field distribution at *EE**′*. From *EE**′* to *FF**′*, the electric field lines are mostly distributed inside the substrate, and two fringing fields exist at the edges of the signal lines, respectively. As the widths of the top and bottom signal lines are changed and become the same, a conventional PSL mode is formed, as shown in field distributions from *FF**′* to *GG**′*.

### 5.3. Cross-Sectional Model and Analytical Formulas

To calculate the characteristic line impedance for cross-sections of the proposed transition, three cross-sectional models are used: a DL-to-CPS transition, a CPS-to-asymmetric PSL transition, and an asymmetric PSL-to-PSL transition. The cross-sectional models for the proposed DL-to-PSL transition are shown in Figure 15. Similar to the DL-to-CPS transition case, the characteristic line impedance of each cross-section corresponding to its electric field distribution can be calculated using the conformal mapping.

For the DL-to-CPS transition section (*AA**′* to *DD**′*), the characteristic line impedance and effective permittivity can be calculated using formulas (6) and (7), respectively.


**CPS-to-Asymmetric PSL Transition**


A cross-sectional model of the CPS-to-asymmetric PSL transition for the conformal mapping is shown in Figure 15a. In [41], analytic formulas of the line capacitances for the transition were derived by the authors’ group. Using the formulas, the capacitances of the analysis regions can be calculated as *C*_5_ (Region V), *C*_6_ (Region VI), and *C*_7_ (Region VII). The characteristic line impedance and effective permittivity of the cross-section of the CPS-to-asymmetric PSL transition can be calculated using formulas (8) and (9), respectively.
(8)$$Z_{0}=\frac{120\pi \varepsilon_{0}}{\sqrt{\varepsilon_{eff}}\left(C_{5}+C_{6}/\varepsilon_{r}+C_{7}\right)}$$
(9)$$\varepsilon_{eff}=\frac{\left(C_{5}+C_{6}+C_{7}\right)}{\left(C_{5}+C_{6}/\varepsilon_{r}+C_{7}\right)}\;$$


**Asymmetric PSL-to-PSL Transition**


A cross-sectional model of the asymmetric PSL-to-PSL transition for the conformal mapping is shown in Figure 15b. The analytic formulas for the line capacitances for the transition were also derived by the authors’ group. Using the formulas, the capacitances of each region can be calculated as *C*_8_ (Region VIII), *C*_9_ (Region IX), and *C*_10_ (Region X) [19]. The characteristic line impedance and effective permittivity of the cross-section of the asymmetric PSL-to-PSL transition can be calculated using formulas (10) and (11), respectively.
(10)$$Z_{0}=\frac{120\pi \varepsilon_{0}}{\sqrt{\varepsilon_{eff}}\left(C_{8}+C_{9}/\varepsilon_{r}+C_{10}/\varepsilon_{r}\right)}$$
(11)$$\;\varepsilon_{eff}=\frac{\left(C_{8}+C_{9}+C_{10}\right)}{\left(C_{8}+C_{9}/\varepsilon_{r}+C_{10}/\varepsilon_{r}\right)}\;$$

### 5.4. Calculation of Characteristic Line Impedance

As an example of the proposed DL-to-PSL transition, a 100 Ω DL connects to a 100 Ω PSL. However, in this case, the CPS impedance is selected as 125 Ω considering a narrow gap width to meet typical PCB fabrication conditions. Therefore, optimal impedance tapering sections from the DL-to-CPS and the CPS-to-PSL are used in the transition. In Figure 16, the calculated line impedance values along the transition using the formulas ((6), (8), and (10)) are compared with the 3D EM-simulated values.

From *BB**′* to *CC**′* in Figure 9, a DL impedance taper is formed as a Klopfenstein impedance taper from 100 Ω (DL) to 125 Ω (narrow-gap DL). In Figure 16a, the characteristic line impedance of the DL-to-narrow-gap DL section is calculated and EM-simulated as a function of gap width *g*. The calculated impedance values deviate from the EM-simulated impedance values by less than 3.7%, where the deviation may have been caused by an overlap between Regions I and III (in Figure 11a), especially when the gap is wide (for *g* > 13 mil).

In the section from *CC**′* to *DD**′* in Figure 9, a narrow-gap DL-to-CPS transition, maintaining the characteristic line impedance of 125 Ω, is designed. The calculated line impedance values as a function of aperture width *s* deviate from the EM-simulated values by less than 4.2%, as shown in Figure 16b, which could be attributed to the insufficient treatment of the boundary between Regions II′ (a) and II′ (b) (in Figure 11c,d).

The section for the CPS-to-asymmetric PSL transition (from *DD**′* to *EE**′* in Figure 9) uses the Klopfenstein impedance taper from a 125 Ω CPS to a 100 Ω asymmetric PSL. The characteristic line impedance in this section is calculated using (8) as a function of the bottom signal linewidth *w* as shown in Figure 16c. The calculated impedance values deviate from the EM-simulated values by less than 3.6%, possibly due to an overlap of Regions VI and VII (in Figure 15a).

Finally, an asymmetric PSL-to-PSL transition (*EE**′* to *FF**′* in Figure 6) is designed to maintain the characteristic line impedance at 100 Ω. The characteristic line impedance in this section is calculated using (10) as a function of the bottom signal linewidth *w* as shown in Figure 16d. The calculated impedance values deviate from the EM-simulated values by less than 3.9%. The calculated values are slightly higher than the EM-simulated values, possibly related to the insufficient consideration of the fringing fields in Regions VIII and IX (in Figure 15b).

## 6. Fabrication and Measurements

The fabricated DL-to-CPS and DL-to-PSL transitions in back-to-back configurations are shown in Figure 17a,b, respectively. The two transitions are fabricated with the 10 mil Duroid 5880 substrate. Each transition consists of two input and two output signal lines, requiring a 4-port vector network analyzer (VNA) to obtain the 4-port S-parameters. The single-ended mode S-parameters can be converted to the mixed-mode S-parameters to verify the performance of the proposed transitions operating in a differential mode [42]. To obtain the 4-port S-parameters with the VNA, a DL-dividing structure, which splits a DL into two separated single-ended microstrip lines, is connected to the transitions. The test structure is designed to maintain the characteristic line impedance of 100 Ω for the differential lines and 50 Ω for the single-ended lines. The overall sizes of the transitions are listed in Table 4.

To connect the transitions under test with the VNA, four end-launch connectors (Southwest Microwave 1092-03A-5, Arizona, Tempe, AZ, USA) are used. For the measurement of the proposed transition, additional auxiliary fixtures, consisting of the DL-dividing structures and the end-launch connectors, are connected to the transition to facilitate the measurement, but the effects of these auxiliary fixtures should be removed from the overall measured results. De-embedding is one of the methods to extract the auxiliary effects from the raw-measured data of the proposed transition in the back-to-back configuration. Various de-embedding techniques were reported, such as TRL (thru-reflect-line) [43], SOL (short-open-load) [44,45], and thru-line de-embedding [46].

Among various de-embedding techniques, a 2X-Thru SFD (Smart Fixture De-embedding) method only requires the data of two directly-connected fixtures, called 2X-Thru data [47,48]. The design topology of the 2X-Thru SFD is shown in Figure 18. With this method, the S-parameters of the 1X-Fixture, which are essential to extract the S-parameters of the DUT, can be obtained from the 2X-Thru data in the time domain and frequency domain [49]. Converting the S-parameters to the T-parameter (transfer scattering parameter) [50], the T-matrix of the DUT can be calculated using (12).
(12)$$\left[T_{DUT}\right]={\left[T_{Fix-left}\right]}^{-1}\times \left[T_{Total}\right]\times {\left[T_{Fix-Right}\right]}^{-1}$$

In order to use the method, the left and right 1X-Fixtures should be symmetric. However, in most cases, an asymmetric error can occur due to manufacturing variations, which require an error correction process. AITT (Advanced Interconnect Test Tool) can provide the 2X-Thru SFD method with error correction techniques for accurate performance extraction [51]. In this paper, in order to obtain an accurate performance of the proposed transitions in the back-to-back configurations, the AITT software is used to remove the auxiliary fixture effects.

Figure 19a,b show pictures of the test set-up for the fabricated transitions in back-to-back configurations, respectively. Figure 19c shows a picture of the fabricated 2X-Thru structure for the 2X-Thru SFD test. Four end-launch connectors are attached to the transition structures under test for the 4-port S-parameter measurements.

Figure 20a,b compare the EM-simulated S-parameters with the de-embedded mixed-mode S-parameters of the proposed transitions in back-to-back configurations with measurements, removing the auxiliary fixture effects from the total structural data including the transitions, the DL-dividing structures, and end-launch connectors. The insertion loss and the return loss of the fabricated DL-to-CPS transition are presented in Figure 20a. It can be observed that the maximum insertion loss of 0.86 dB per transition is obtained up to 40 GHz, and the return loss of the transition is more than 10 dB from DC to 40 GHz. For the fabricated DL-to-PSL transition, the insertion loss and return loss are shown in Figure 20b. The transition provides the maximum insertion loss of 1 dB per transition from DC to 35.6 GHz, and the maximum insertion loss of 1.34 dB per transition from 35.6 to 40 GHz with the return loss of greater than 10 dB from DC to 40 GHz. The measured insertion losses of the transitions are slightly more than the EM-simulated results. The measured return losses of the transitions have some deviations as compared with the simulation results above 12 GHz but are still greater than 10 dB.

Therefore, both of the two proposed DL-to-BL transitions, i.e., DL-to-CPS and DL-to-PSL transitions, are proved to provide ultra-wideband frequency bandwidth of several tens of GHz, supporting digital transmission rates of over 200 Gbps per lane in PCBs [52]. In addition, the proposed transitions are compatible with the conventional DL-based printed circuit boards and take much less layout space, while providing multiple advantages of using BLs in terms of ultra-wide bandwidth, common-mode rejection, phase recovery, reduced crosstalk, and skew reduction. Furthermore, by utilizing the proposed transitions, BL-based common mode rejection filters, equalizers, and other components for super-high-speed transmission may also be introduced into the DL-based PCBs.

## 7. Conclusions

In order to support super-high-speed digital data transmission on printed circuit boards, two ultra-wideband DL-to-BL transitions, i.e., DL-to-CPS and DL-to-PSL transitions, are presented in this paper. A conventional digital transmission line (DL) has a fundamental limitation on the maximum frequency bandwidth due to skew generation and crosstalk, which generate EM interference. The balanced lines (BLs) inherently provide ultra-wideband performance with multiple advantages. Since the transitional structures from the DL to BLs typically dominate the frequency bandwidth, the proposed DL-to-BL transitions are designed to provide optimal performance with the compatibility of DL-based PCBs.

In designing the transitions, the characteristic line impedances of the cross-sections of the transition are accurately and efficiently calculated using the analytical formulas based on conformal mapping. Two signal lines and the bottom ground line of the transitions are adjusted to form an optimal impedance taper to attain the ultra-wideband performance. The DL-to-CPS transition connecting a 100 Ω DL and a 138 Ω CPS, and the DL-to-PSL transition connecting a 100 Ω DL and a 100 Ω PSL are designed and fabricated to demonstrate the performance of the proposed transitions. The implemented DL-to-CPS and DL-to-PSL transitions both provide high-quality signal performance from DC to over 40 GHz.

Therefore, the proposed DL-to-BL transitions can be applied to achieve super-high-speed digital data transmission with over 40 GHz bandwidth, which is more than four times the bandwidth of the DL, supporting over 200 Gbps of digital data transmission on PCBs, for the next generation advanced communications.

## Figures and Tables

**Figure 1 sensors-22-06873-f001:**
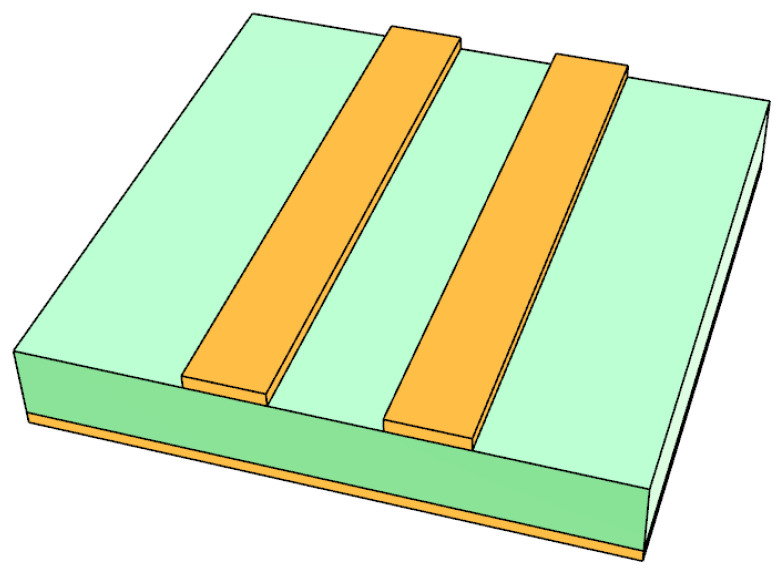
A perspective view of a conventional DL.

**Figure 2 sensors-22-06873-f002:**
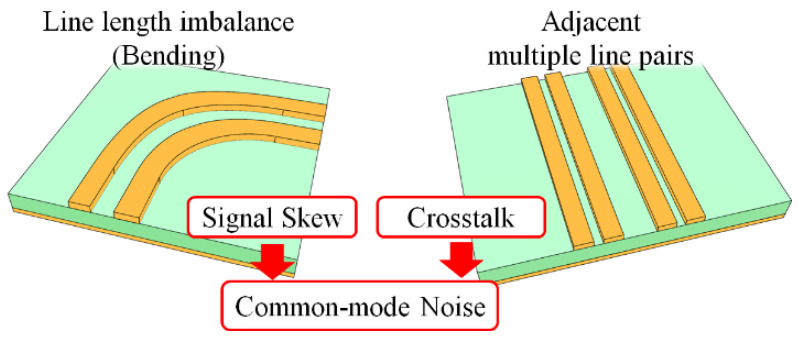
Examples of complex structures in a high-speed circuit board causing the common-mode noise.

**Figure 3 sensors-22-06873-f003:**
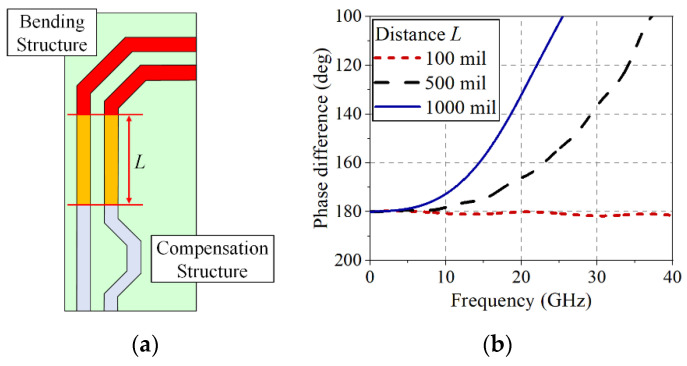
(**a**) Compensation structure located at a distance (*L*) from the bent DL, (**b**) phase difference between the two DL lines after applying the compensation structure (path difference of 30 mil, DL linewidth of 10 mil, and gap width of 5 mil with a 5 mil FR-4 substrate).

**Figure 4 sensors-22-06873-f004:**
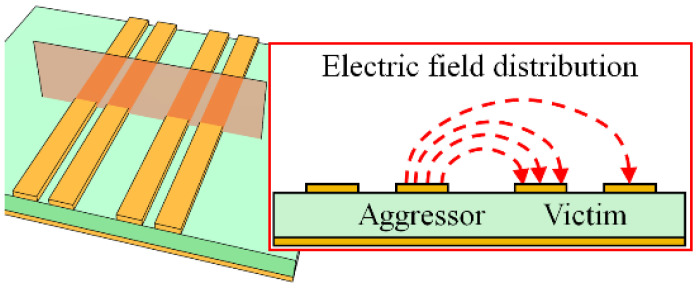
Crosstalk between the DL pairs, asymmetrically affecting each line of the DL.

**Figure 5 sensors-22-06873-f005:**
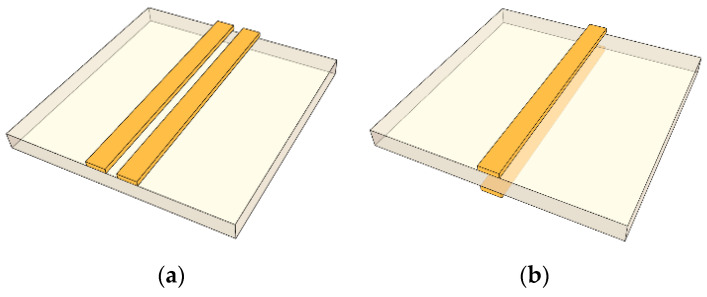
Perspective views of the balanced lines: (**a**) CPS, (**b**) PSL.

**Figure 6 sensors-22-06873-f006:**
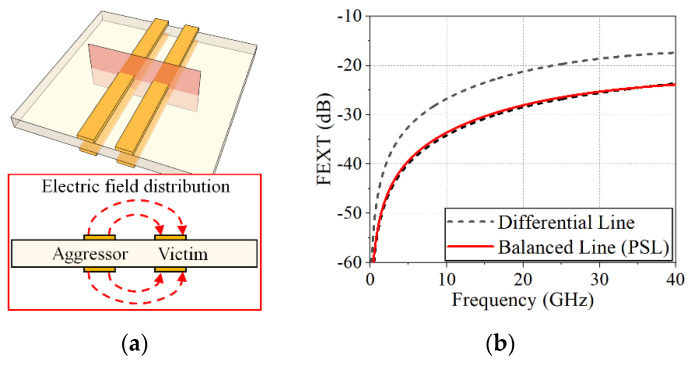
(**a**) Crosstalk level between the PSL pairs, (**b**) crosstalk levels due to adjacent lines (DL pair: linewidth of 10 mil, gap width of 10 mil, distance of 30 mil with a 5 mil FR-4 substrate, PSL pair: width of 10 mil, distance of 30 mil with a 5 mil FR-4 substrate).

**Figure 7 sensors-22-06873-f007:**
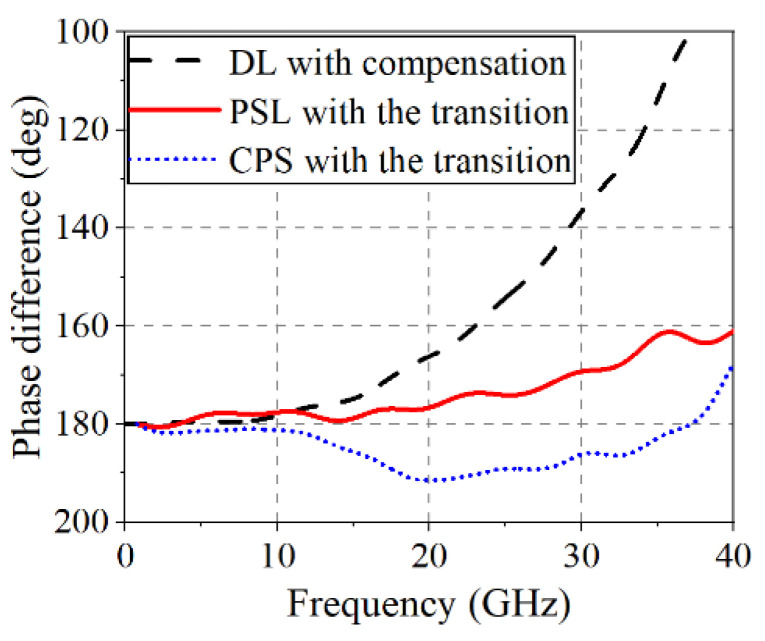
Phase differences between the two lines of the DL and the PSL in the presence of a 30 mil DL line imbalance (DL: DL linewidth of 10 mil, gap width of 5 mil, and distance L of 500 mil with a 5 mil FR-4 substrate, CPS: CPS linewidth of 10 mil, and gap width of 5 mil with a 5 mil FR-4 substrate, PSL: PSL linewidth of 5 mil with a 5 mil FR-4 substrate).

**Figure 8 sensors-22-06873-f008:**
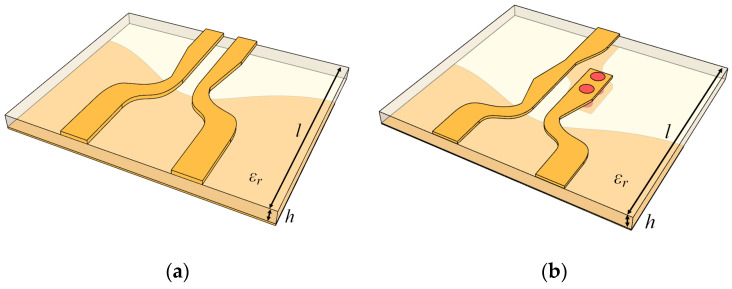
Perspective views of the proposed DL-to-BL transitions: (**a**) DL-to-CPS transition, (**b**) DL-to-PSL transition.

**Figure 9 sensors-22-06873-f009:**
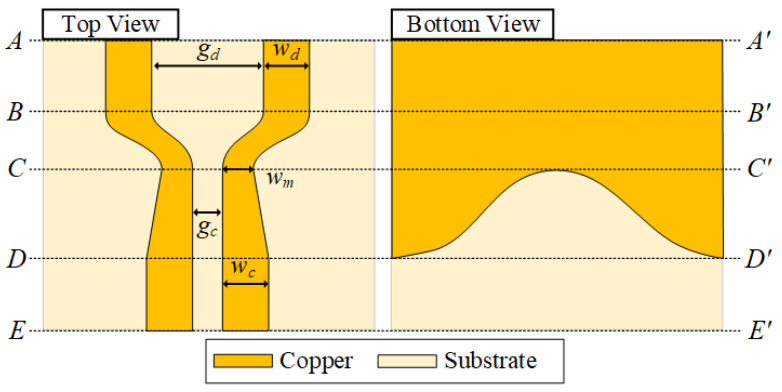
Top and bottom views of the proposed DL-to-CPS Transition.

**Figure 10 sensors-22-06873-f010:**
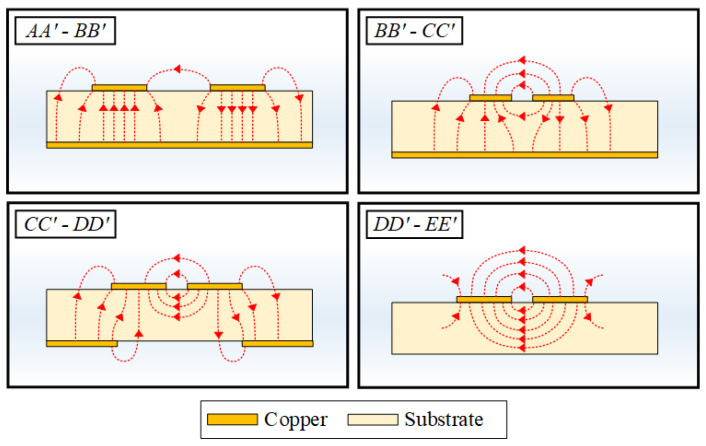
Electric field distributions at the cross-sectional stages of the proposed transition.

**Figure 11 sensors-22-06873-f011:**
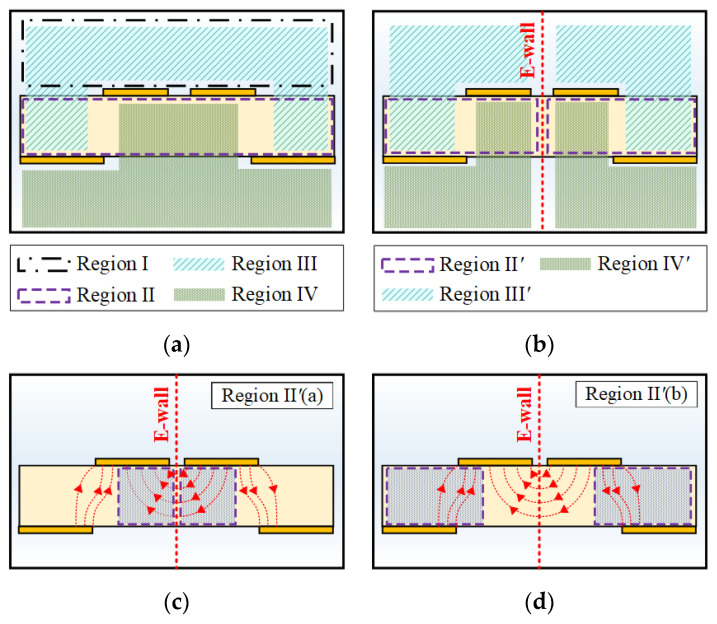
Cross-sectional model of the DL-to-CPS transition: (**a**) four analysis regions, (**b**) analysis regions divided by an E-wall, located in the middle of the transition, (**c**) Region II′(a) between the signal line and the E-wall, and (**d**) Region II′(b) between the signal line and the ground line.

**Figure 12 sensors-22-06873-f012:**
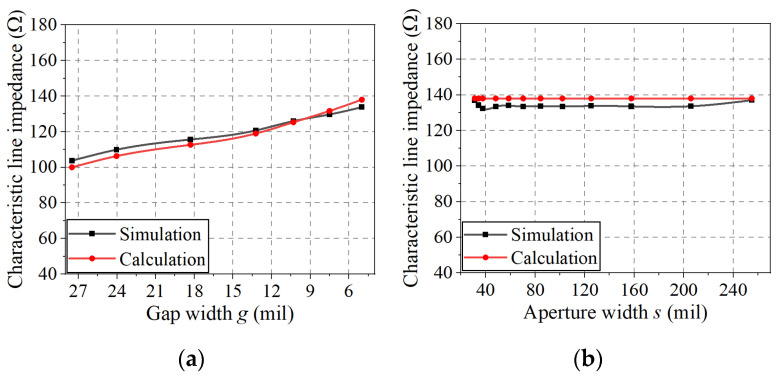
Calculated and EM-simulated characteristic line impedances of the DL-to-CPS transition: (**a**) *BB**′* to *CC**′*, (**b**) *CC**′* to *DD**′*.

**Figure 13 sensors-22-06873-f013:**
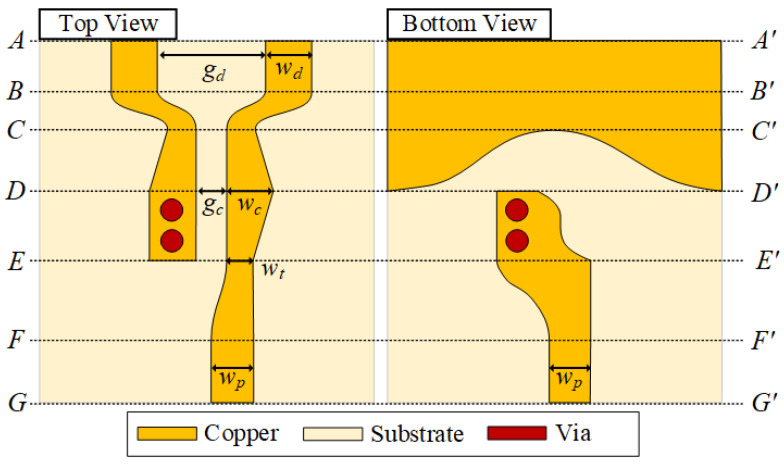
Top and bottom views of the proposed DL-to-PSL transition.

**Figure 14 sensors-22-06873-f014:**
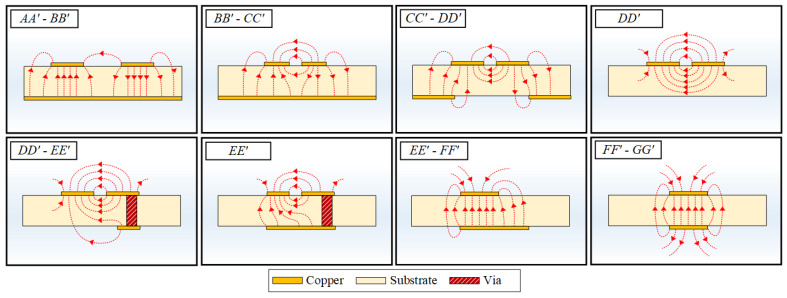
Electric field distributions of the cross-sectional stages of the DL-to-PSL transition.

**Figure 15 sensors-22-06873-f015:**
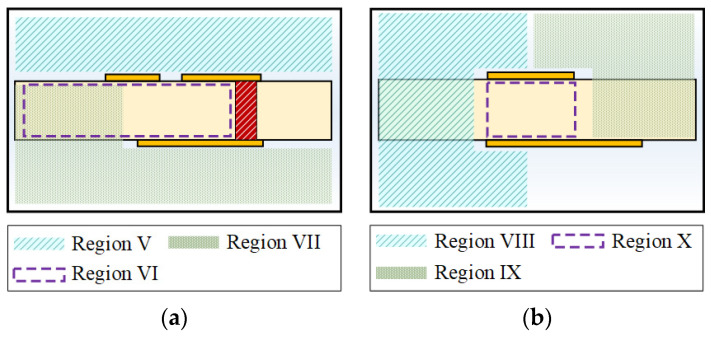
Analysis regions of the cross-section of the CPS-to-PSL transition: (**a**) for the CPS-to-asymmetric PSL transition, (**b**) for the asymmetric PSL-to-PSL transition.

**Figure 16 sensors-22-06873-f016:**
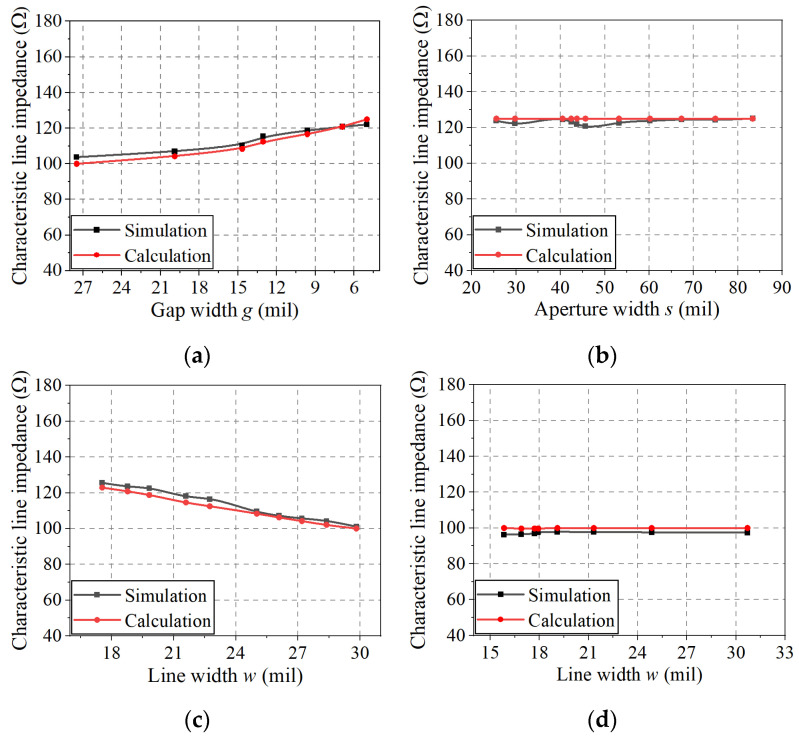
Calculated and EM-simulated characteristic line impedances of the proposed DL-to-PSL transition: (**a**) *BB**′* to *CC**′*, (**b**) *CC**′* to *DD**′*, (**c**) *DD**′* to *EE**′*, (**d**) *EE**′* to *FF**′*.

**Figure 17 sensors-22-06873-f017:**
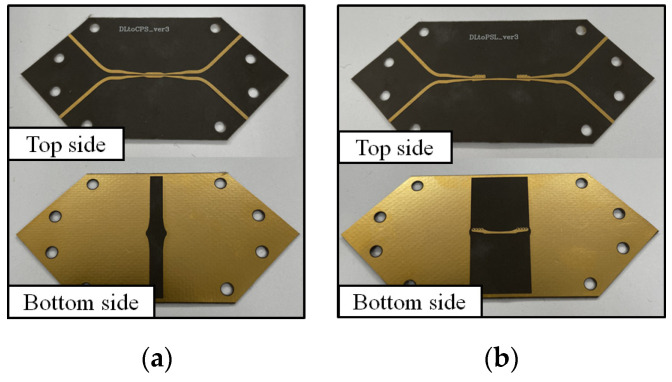
Fabricated transitions in back-to-back configurations: (**a**) DL-to-CPS transition, (**b**) DL-to-PSL transition.

**Figure 18 sensors-22-06873-f018:**
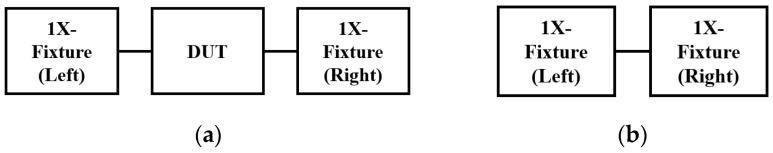
Design topology of the 2X-Thru SFD method: (**a**) whole structure consisting of the DUT and symmetric input/output auxiliary structures, (**b**) 2X-Thru structure.

**Figure 19 sensors-22-06873-f019:**
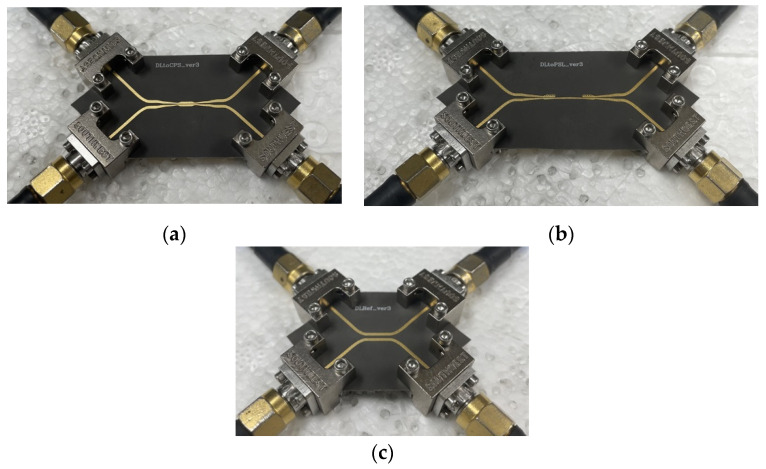
Pictures of the fabricated transitions and the Thru structure with the end-launch connectors: (**a**) DL-to-CPS transition, (**b**) DL-to-PSL transition, (**c**) 2X-Thru structure.

**Figure 20 sensors-22-06873-f020:**
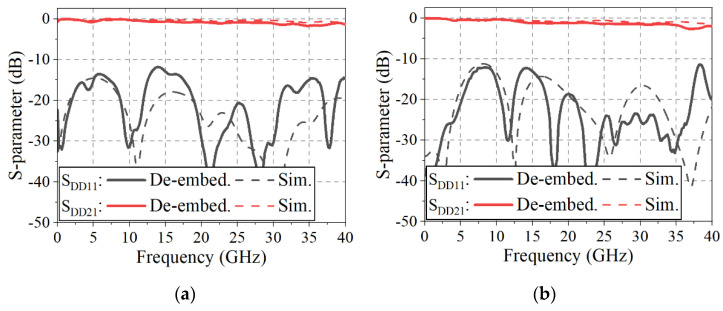
EM–simulated and measured S–parameters of the proposed transitions: (**a**) DL–to–CPS transition, (**b**) DL–to–PSL transition.

**Table 1 sensors-22-06873-t001:** Comparison of the ultra-wideband transitions.

Ref.	Structure	Characteristic Line Impedance [Ω]	Operating Bandwidth [GHz]
[19]	CPS-to-PSL	147/50, 120/50	6.4–40
[20]	CPS-to-MSL	147/50	6–40
[21]	PSL-to-MSL	50/50	DC-40

**Table 2 sensors-22-06873-t002:** Design parameters of the DL-to-CPS transition.

Parameters	*w_d_*	*w_c_*	*w_m_*	*g_d_*	*g_c_*	*l*
**Size in mil** **(mm)**	26.8 (0.68)	20 (0.51)	9.78 (0.25)	27.5 (0.70)	5 (0.13)	230 (5.84)

**Table 3 sensors-22-06873-t003:** Design parameters of the DL-to-PSL transition.

Parameters	*w_d_*	*w_c_*	*w_t_*	*w_p_*	*g_d_*	*g_c_*	*l*
**Size in mil** **(mm)**	26.8 (0.68)	20 (0.51)	11 (0.28)	15. 9(0.40)	27.5 (0.70)	5 (0.13)	410 (5.84)

**Table 4 sensors-22-06873-t004:** Overall sizes of the fabricated DL-to-BL transitions.

Transition	Size [mil (mm)] *
DL-to-CPS	660 × 1000 × 11.4 (16.8 × 25.4 × 0.29)
DL-to-PSL	970 × 1000 × 11.4 (24.6 × 25.4 × 0.29)

* Transition dimensions: (length in transverse axis) × (length in longitudinal axis) × (height of the transition).

## Data Availability

Not applicable.

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
