# Peer review of "Ultra-Wideband Differential Line-to-Balanced Line Transitions for Super-High-Speed Digital Transmission"

_sensors, 2022, doi:10.3390/s22186873_

Round 1
Reviewer 1 Report
In this paper, the authors propose two categories of transmission structures for digital signals to support super-high-speed digital data transmission, especially for beyond 5G communications. The proposed arrangement involves the use of balanced lines (BLs) through ultra-wideband transition structures connecting from the DL. Although the proposed concept is interesting, there are a few points to clarify and some comments that need to be considered:
-How the substrate characteristics were chosen (thick, permittivity...)?
-What about the equivalent circuit model of the proposed transitions?
-About the repeatability of the transitions: the reviewer would like to see some more measurements to confirm that is really repeatable. For example, the authors can just unmount the test fixture module, reassemble again a couple of times and measure the S-parameters.
-It is interesting to compare the work with the current state-of-the-art approaches from the literature.
-In Figure 19, SDD11 and SDD21 must be written in accordance with the mathematical presentation of the S-parameters (DD11 and DD21 in smaller sizes)
-The English usage including grammatical errors must be checked throughout the manuscript. Please, avoid using excessively long sentences and repetition in the abstract, introduction, and conclusion. In addition, the abstract is too long and can be reduced.
Reviewer 2 Report
The author has to add the comparison table of the past work.
What is the innovation in your method?
What is the correlation between Phase difference and frequency?
Is there any magnetic field effect on your design?
What is the correlation between the magnetic and electric fields in your design?
As per your figure 8, is there any coupling effect on the design? What type of coupling effect is in your design?
What is the correlation between the different types of substrates in your design?
Is there any effect on the Er with frequency?
The author has to add the different types of copper property.
What is the difference between the Calculation of Characteristic Line Impedance and simulation line impedance value?
What is the practical application of your design?
The author has to add the DUT test image for more clarification of the test.
overall paper is good
Reviewer 3 Report
The paper presents two different connecting structures to transform a differential transmission line (DL) to a balanced transmission line (BL), either a parallel stripline or a coplanar stripline. The paper is well written and gives a good background of the current scenario where most of high-speed digital chips and interfaces are designed to mate a DL. The authors summarize some of the advantages of BLs as a counterpart of the DLs for use in very-high speed digital circuit boards, which justifies the need of connecting structures between BL and DL.
I have a few comments I would like to share with the authors.
1.) The introduction is well written and it provides enough information to understand your proposal and its advantages. However, some statements should be better supported with appropriate references, for example:
50 Thereby, differential signaling is typically used to mitigate the external noise and to detect 50 the digital signal of low voltage
57 That is why the DL is widely adopted, over the single-ended 57 (microstrip) line, for the high-speed signal lines in most of the digital circuits.
89 In addition, the two strongly-coupled lines in a BL tend to self-recover 89 the phase imbalance related to a skew. One of the main reasons for not popularly using 90 BLs in digital circuit boards until now is the absence of practical transitional structures 91 between the commonly-used DL and the BLs.
2.) The authors claim that BLs support 10s to 100s Gbps several times throughout the manuscript, some references should be provided to support this statement.
3.) I believe lines 161 to 167 are inherited from the manuscript template. In that is the case, they must be deleted.
4.) Line 187. I think the following statement should be rewritten. Instead of: “ and the PSL is an antipodal structure that a dielectric substrate is placed between the two conductor lines” It should be: “and the PSL is an antipodal structure <<where>> a dielectric substrate is placed between the two conductor lines”. Although I am not sure if the current form is wrong, my proposal sounds much better to me.
5.) Could you explain the “phase-recovering property” of the BLs? How does it work? Could you include some formula or references to support this statement?
6.) The authors claim that PSL is immune to skewing and I agree with them because of its geometry. However, after explaining this, you automatically claim that BLs are immune to skewing, could you explain why CPS is also immune to skewing?
7.) I find the PSL really interesting, however you do not mention one of the disadvantages of using this transmission line, which is the PCB layers misalignment, which modifies the ideal geometry. Have you modeled this misalignment? Have you calculated, simulated or measured how it affects your design? Is it negligible? Or should it be taken into account as a common mode noise source?
8.) Figure 13. I believe it is not to scale, I can tell Wp is visually different for the top and bottom views, please redraw or rescale it to correct the issue, at least to a point where the eye can’t tell the difference.
9.) What is the thickness of the board? Is it 0.29mm?
